# Turbo4DGen: Ultra-Fast Acceleration for 4D Generation

Yuanbin Man[1]  Ying Huang[1]  Zhile Ren[2]  Miao Yin[1]

## Abstract

4D generation, or dynamic 3D content generation, integrates spatial, temporal, and view dimensions to model realistic dynamic scenes, playing a foundational role in advancing world models and physical AI. However, maintaining long-chain consistency across both frames and viewpoints through the unique spatio-camera-motion (SCM) attention mechanism introduces substantial computational and memory overhead, often leading to out-of-memory (OOM) failures and prohibitive generation times. To address these challenges, we propose Turbo4DGen, an ultra-fast acceleration framework for diffusion-based multi-view 4D content generation. Turbo4DGen introduces a spatiotemporal cache mechanism that persistently reuses intermediate attention across denoising steps, combined with dynamically semantic-aware attention pruning and an adaptive SCM chain bypass scheduler, to drastically reduce redundant SCM attention computation. Our experimental results show that Turbo4DGen achieves an average 9.7× speedup without quality degradation on the ObjaverseDy and Consistent4D datasets. The project page is available at https://noodle-lab.github.io/turbo4dgen/.

## 1. Introduction

4D generation, also known as dynamic 3D content generation, has gained increasing attention following the remarkable success of diffusion models. It is crucial for real-world applications, such as AR/VR and immersive content creation. More importantly, with the capability of capturing spatiotemporal dynamics, 4D generation serves as a foundation for world modeling, which enables intelligent agents to plan and interact within virtual or physical environments. Recent state-of-the-art 4D generation methods, including

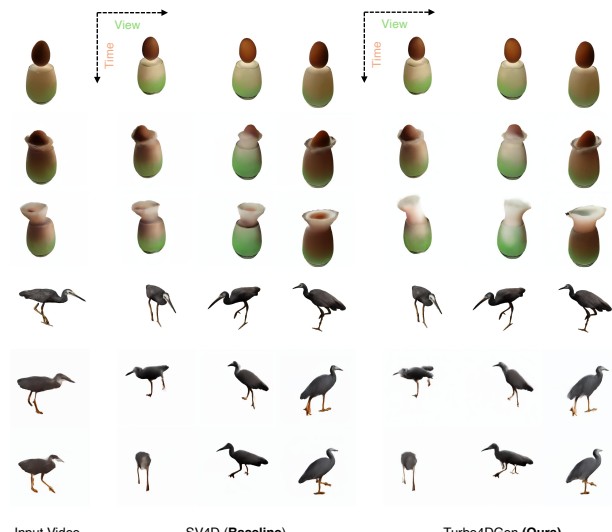

*Figure 1.* Generated examples of Turbo4DGen, in comparison with the baseline, SV4D (Xie et al., 2025). Our Turbo4DGen completes the above 4D generation examples in only **9.78s** and **12.15s**, respectively, whereas SV4D requires around 2 mins (110.85s and 118.76s), yielding **11.33×** and **9.77×** speedups without sacrificing content quality.

SV4D (Xie et al., 2025), Consistent4D (Jiang et al., 2024), 4Real-Video (Wang et al., 2025), and CAT4D (Wu et al., 2025), can generate multi-second, multi-view 4D videos from a single monocular input.

However, deploying a 4D generation model on a modern GPU is computationally expensive, especially when generating high-resolution 4D videos. For example, generating a 768×768, 21-frame, 8-view, 2-second 4D content using the existing model, e.g., SV4D (Xie et al., 2025), on a single NVIDIA RTX 6000 Ada GPU requires more than 2 minutes. Such intensive computation needs have significantly limited the practicality of 4D generation, further hindering the advancements of world modeling and physical AI. Technically, 4D generation is inherently complex due to the rigorous requirements for spatial, temporal, and view consistency. Unlike image generation, e.g., StableDiffusion (Rombach et al., 2022), which focuses solely on spatial content within a single frame and has no temporal or consistency constraints, or video generation, e.g., StableVideoDiffusion (Blattmann et al., 2023a), which requires only frame-wise temporal computation, 4D generation invokes higher-dimensional

[1]Department of Computer Science and Engineering, University of Texas at Arlington, TX, USA [2]Independent Researcher, WA, USA. Correspondence to: Miao Yin <miao.yin@uta.edu>.

*Proceedings of the 43rd International Conference on Machine Learning*, Seoul, South Korea. PMLR 306, 2026. Copyright 2026 by the author(s).

computations across spatio-camera-motion (SCM) space, making accelerating 4D generation significantly more demanding and challenging.

Unfortunately, there are no existing works to resolve the computational problems in 4D generation. Although some approaches have been developed for diffusion-based models, they are limited to 2D generation, i.e., image tasks. For example, DeepCache (Ma et al., 2024) accelerates image diffusion models by exploiting state redundancy across sequential denoising steps, caching and reusing features at the block level to reduce computations. However, 4D generation models invoke significantly more complex blocks with unique spatio-camera-motion attention chains, making such caching strategies unrealistic to apply directly. In contrast, AT-EDM (Wang et al., 2024b) applies single-denoising-step token pruning with a denoising-step-aware scheduler to iteratively reduce computational cost. Although effective for a single attention block, these approaches are not applicable to SCM attention chains.

In this paper, we propose Turbo4DGen, an ultra-fast acceleration framework for 4D generation, as shown in Figure 1. Turbo4DGen identifies and removes the redundant computation-intensive operations at multi-scale attention granularity, i.e., token level, block level, and chain level, based on the designed rolling cache and adaptive bypassing mechanism. The proposed Turbo4DGen are motivated by three key observations on the SCM attention, which account for the major latency and memory usage during both individual denoising steps and the overall generation process. ① A noticeable similarity among the SCM attention outputs exists across different denoising steps. ② The semantic importance of the token derived from the spatial block can serve to guide the pruning of subsequent camera and motion blocks within the same denoising step. ③ 4D generation exhibits highly dynamic redundancy patterns, which the attention similarity across diffusion steps can monitor.

Overall, the contributions of our proposed Turbo4DGen are summarized as follows:

- We systematically analyze the SCM attention chain, which primarily results in the computation and memory overhead, in the 4D generation process. Based on our analysis and observations, we identify the main challenges that hinder the direct application of existing image generation acceleration methods, as well as the unique properties that can be leveraged to reduce redundant computations.

- We propose, for the first time, an ultra-fast acceleration framework for 4D generation. Specifically, Turbo4DGen skips SCM *block-level* computations through a designed rolling cache mechanism, according to which a semantic-aware pruning approach iden-

tifies *token-level* redundancy and refills the attention without impairing semantic-spatiotemporal coherence. Furthermore, we propose a bypass scheduler that can adaptively skip the entire *chain-level* SCM attention computations.

- We comprehensively evaluate Turbo4DGen on the Consistent4D (Jiang et al., 2024) and Objaverse (Deitke et al., 2022) datasets from both quantitative and qualitative perspectives. Experimental results show that Turbo4DGen achieves **an average of 9.7×** **speedup and 24.2% memory reduction** without quality degradation compared to the baseline.

## 2. Background of 4D Generation

In this section, we will briefly introduce the 4D scene generation process and model architectures. Generally, 4D generation methods, e.g., Diffusion4D (Liang et al., 2024a), DreamFusion (Poole et al., 2022), 4Dfy (Bahmani et al., 2024), employ the score distillation sampling (SDS) loss to optimize 4D scene synthesis from 2D diffusion models. To improve 4D generation quality, state-of-the-art methods such as SV4D (Xie et al., 2025) and its improved variant SV4D 2.0 (Yao et al., 2025) leverage the robust 3D generation model SV3D (Voleti et al., 2024) as a prior and utilize the SCM attention chain to align spatial-temporal-view consistency. Other works, such as 4Real (Yu et al., 2024b) and CAT4D (Wu et al., 2025), also follow this paradigm. Without loss of generality, we next use SV4D (Xie et al., 2025), the only open-source 4D generation method, to detail the generation process.

**4D Generation Objective.** Given a monocular video $J \in \mathbb{R}^{F \times D}$ of a dynamic object with a sequence of $F$ dynamic frames and the merged image dimension $D$, 4D generation aims to generate consistent multi-view video $M \in \mathbb{R}^{V \times F \times D}$ at $V$ views conditioning on camera trajectory $\pi = \{(e_v, a_v)\}_{v=1}^V \in \mathbb{R}^{V \times 2}$, where $e_v$ and $a_v$ are elevation and azimuth angles relative to the input view of the monocular video.

**Multi-View Video Diffusion.** Typically, multi-view videos are generated from noise by gradually reversing a forward noising process. Specifically, the forward-noising process first encodes the multi-view video into a latent representation $Z^0$ using a VAE encoder. Then it generates the noisy latent $Z^t \in \mathbb{R}^{F \times V \times H \times W \times C}$ by adding Gaussian noise into the clean latent $Z_0$ as follows:

$$q(Z^t \mid Z^0) = \mathcal{N}(Z^t; \alpha_t Z^0, \beta_t^2 \mathbf{I}), \tag{1}$$

where $\alpha_t$ and $\beta_t$ are timestep-dependent constants with $t$ uniformly sampled from $\{1, \ldots, T\}$, $H$, $W$ and $C$ denote the image height, width, and channel. During the reverse denoising process, the denoising network $\mathcal{G}_\theta$ is trained to

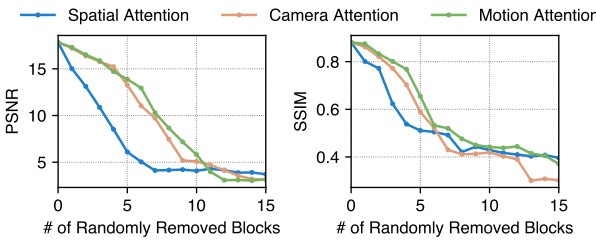

*Figure 2.* Performance analysis of removing spatial, camera, or motion attention blocks. It is observed that *spatial attention block* plays a more critical role in the SCM attention chain.

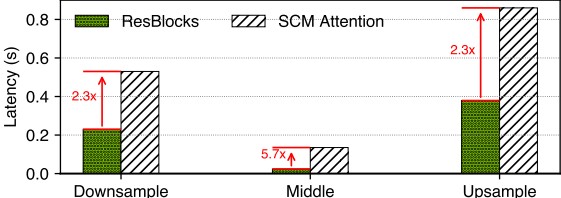

*Figure 3.* Latency analysis of multiple components in 4D generation (Xie et al., 2025). The results show that SCM attention is the main bottleneck, accounting for most of the computational overhead.

progressively predict the clean version of $\boldsymbol{Z}^t$ via multiple denoising steps. The corresponding training objective of $\mathcal{G}_\theta$ can be simplified as:

$$L(\theta) = \mathbb{E}_{\mathcal{G}, \boldsymbol{Z}^t, \boldsymbol{J}, \boldsymbol{\pi}, t} \left[ \left\| \boldsymbol{Z}^0 - \mathcal{G}_\theta(\boldsymbol{Z}^t, \boldsymbol{J}, \boldsymbol{\pi}, t) \right\|_2^2 \right]. \quad (2)$$

In the inference process, the multi-view video diffusion model takes the noise sampled from a standard Gaussian distribution as input and iteratively generates a sequence of cleaner multi-view videos.

**Spatial-Camera-Motion (SCM) Attention Mechanism.** The multi-view video denoising network $\mathcal{G}_\theta$ is a UNet-like structure consisting of multiple 3DConv layers followed by an SCM attention chain (Yao et al., 2025), $\mathcal{F}_{\text{SCM}} = \{\mathcal{F}_s, \mathcal{F}_c, \mathcal{F}_m\}$. The spatial attention block $\mathcal{F}_s$ captures image-level details by performing attention across the spatial axes, i.e., $H, W$. For multi-view consistency, the camera attention block $\mathcal{F}_c$ then transposes the features and computes attention across the multi-view axes, i.e., $V$. Finally, the motion attention block $\mathcal{F}_m$ applies the attention along the video frame dimension, i.e., $F$. Formally, given the input $\boldsymbol{Z}$ and prior $\boldsymbol{K}$ at the denoising step $t$, the forward process within block $\mathcal{F}$ can be defined as:

$$\mathcal{F}^t(\boldsymbol{Z}^t, \boldsymbol{K}) = \text{FFN}(\boldsymbol{Z}^t + \text{Attn}(\boldsymbol{Z}^t, \boldsymbol{K}_s)). \quad (3)$$

Here, $\text{FFN}(\cdot)$ and $\text{Attn}(\cdot, \cdot)$ denote the feed-forward and attention computations. Note that each of the spatial $\mathcal{F}_s$, camera $\mathcal{F}_c$, and motion $\mathcal{F}_m$ blocks involves complex attention operations. Overall, the output of SCM chain $\boldsymbol{A}^t \in \mathbb{R}^{F \times V \times H \times W \times C}$ is obtained by

$$\boldsymbol{A}^t = \mathcal{F}_m^t \big( \mathcal{F}_c^t \big( \mathcal{F}_s^t (\boldsymbol{Z}^t, \boldsymbol{K}_s), \boldsymbol{K}_c \big), \boldsymbol{K}_m \big), \quad (4)$$

where $\boldsymbol{K}_s \in \mathbb{R}^{F \times V \times 1 \times C}$, $\boldsymbol{K}_c \in \mathbb{R}^{F \times H \times W \times C}$, $\boldsymbol{K}_m \in \mathbb{R}^{V \times H \times W \times C}$ represent the spatial, multi-views and multi-frames priors (Radford et al., 2021) extracted from input video, respectively.

## 3. Challenges & Motivations

In this section, we present the main challenges in accelerating 4D generation and the corresponding opportunities, motivating us to propose Turbo4DGen.

**Challenge 1: Complex inter-block connection invalidates simple cache design.** As shown in Figure 3, the SCM attention chain, which consists of spatial, camera, and motion attention blocks, dominates the computational overhead due to the large latent and context shapes required for multi-view video generation. A naive solution is to cache the final this-step SCM output to skip the entire SCM computation for the next step. Unfortunately, due to the complex residual connection design across blocks and the absence of next-step inter-block computation, the generation quality is significantly degraded, as shown in Figure 2. Furthermore, in 4D generation, there are considerably more denoising steps than image tasks, resulting in a non-negligible token selection overhead, which leads to non-realistic per-step token selection with existing sparse attention techniques (Sheng et al., 2023; Zhang et al., 2023; 2025; Gao et al., 2024; Desai et al., 2025).

**Opportunity: Significant inter-step SCM attention similarity exists in the denoising process.** Fortunately, we observe that the block-level attention outputs exhibit high similarity (see Figure 4) between two denoising steps, even with a step interval of 2. Moreover, the similarity tends to be higher during the later denoising stages. In contrast, other Conv3D blocks within the UNet-like structure demonstrate much lower similarity throughout the denoising process. This observation motivates us to propose a block-level caching mechanism to efficiently retrieve and reuse attention maps from each type of SCM blocks before denoising steps, thereby significantly reducing this-step attention computations.

**Challenge 2: Spatiotemporal inconsistency in sparsifying SCM attention.** Naively, to accelerate diffusion-based models, existing approaches, e.g., AT-EDM (Wang et al., 2024b), apply pruning mechanisms to individual attention blocks. However, this strategy is only applicable to single-attention diffusion models. In contrast, 4D generation models rely on the SCM attention chain to ensure spatiotemporal consistency. Sequentially applying those pruning methods to the spatial, camera, and motion attention blocks can disrupt this consistency and degrade results. Therefore, to accelerate the entire SCM attention chain and preserve the spatiotem-

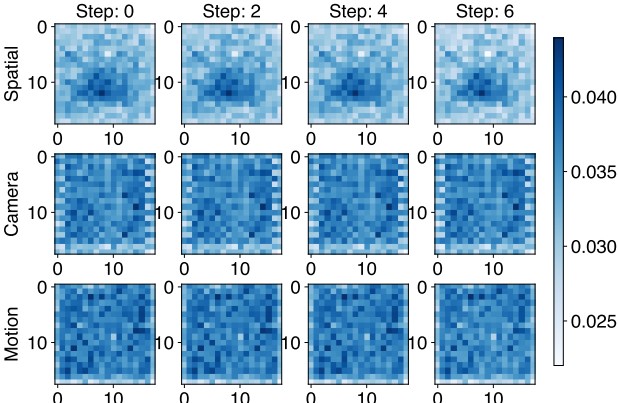

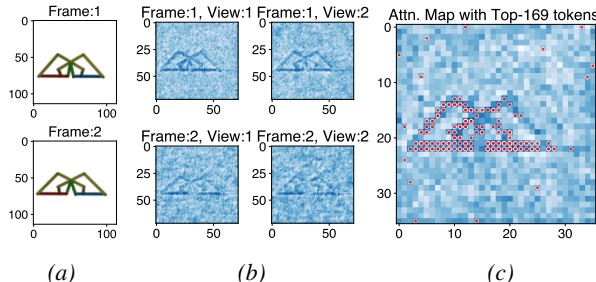

*Figure 5.* Visualization of the spatial cross-attention map indicating semantic representations. (a) Two frames from a reference video; (b) The corresponding spatial cross-attention map; (c) Top-$K$ relevant tokens (dotted in red) representing semantic features.

*Figure 4.* The outputs of the SCM attention blocks in the final layer (downsampled to $16 \times 16$ for visualization) across adjacent denoising steps (sampling every two steps) exhibit a cosine similarity exceeding 95%, indicating strong redundancy between consecutive steps.

poral consistency, a unique long-chain attention sparsifying approach is necessary to address this challenge. This unique long-chain SCM attention mechanism also causes constant-value filling of pruned attentions, while effective in image generation tasks, a significant degradation of 4D generation quality, as shown in Figure 6b.

**Opportunity: Intra-step spatial cross-attention can guide sparsifying camera and motion attention.** In Figure 2, we have observed that spatial attention is most important to the final generation quality. Figure 5 further illustrates that the cross-attention map generated by the spatial block can identify the key tokens corresponding to generated objects, which can guide pruning in subsequent camera and motion attention blocks to preserve those object-essential tokens. By doing so, we can identify the essential tokens in the spatial attention block with low computational complexity. Thus, only the identified tokens (compressed along the $H$ and $W$ axes) need to be included in the computation of camera and motion attention, which are more computationally intensive than spatial attention, thereby significantly reducing latency. More fortunately, with the aforementioned block-level attention cache, we can refill the pruned location with cached values, avoiding the degraded generation quality caused by naive constant-value assignments.

**Challenge 3: 4D generation exhibits a highly dynamic redundancy pattern across SCM attention chains.** Building upon the similarity observed across adjacent steps of the diffusion model, several studies, e.g., DeepCache (Ma et al., 2024), have proposed skipping UNet blocks at certain steps to reduce latency. However, 4D generation exhibits a different redundancy pattern. As shown in Figure 7, the redundancy is highly dynamic across steps, remaining low and unstable during the early denoising steps but increasing significantly in the later steps. Consequently, instead

of bypassing blocks at fixed steps, a dynamic redundancy-removal mechanism is required to accelerate 4D generation by adaptively skipping blocks across varying denoising steps.

**Opportunity: Attention similarity across steps reflects the degree of redundancy.** Figure 7 illustrates the trend of average similarity rate $V_{\text{ASR}}$ during the generation process, defined as the average similarity of all SCM caches throughout denoising steps. As the denoising steps proceed, $V_{\text{ASR}}$ gradually increases and remains at a high value, consistent with the trend in the output similarity of the m-to-last chain. Therefore, $V_{\text{ASR}}$ can reflect the stability of denoising features, which is sufficient to serve as a reliable indicator for dynamically scheduling chain usage in subsequent denoising steps.

## 4. Methodology: Turbo4DGen

In this section, we present the detailed designs of Turbo4DGen, comprising three key components.

### 4.1. Overview

The overview of the proposed Turbo4DGen is illustrated in Figure 8. Specifically, Turbo4DGen accelerates 4D generation by reducing redundant computations at multi-scale granularity in the SCM attention mechanism across block, token, and chain levels.

*At the block level,* building upon the observation that the outputs of attention blocks exhibit similarities across different denoising steps, we propose a *block attention reuse mechanism with rolling cache*. Turbo4DGen independently cache the attention outputs of spatial, camera, and motion blocks into the rolling cache at denoising steps, and reuse them in the subsequent denoising step.

*At the token level,* we propose a *semantic-aware and cache-enhanced token pruning* technique to reduce intra-block attention computations. Turbo4DGen estimates each token's

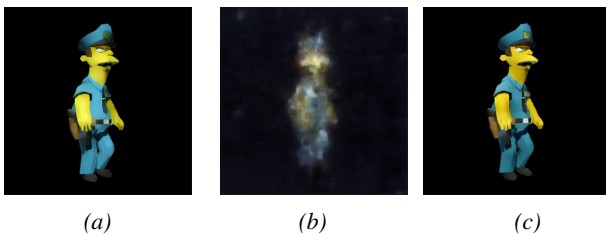

*(a)*       *(b)*       *(c)*

*Figure 6.* Visualization of a generated frame example with different refilling methods after token pruning. (a) Baseline; (b) Pruning with zero-value refilling; (c) Pruning with block cache refilling.

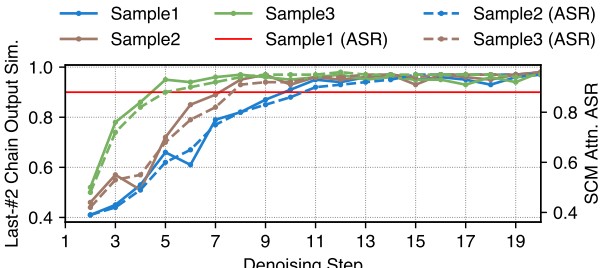

*Figure 7.* Three randomly selected samples from the ObjaverseDy (Deitke et al., 2022) dataset for the baseline. The similarity of the output from the last-#2 chain at steps $t$ and $t - 1$ is computed, along with the SCM average similarity rate (ASR) $V_{\text{ASR}}$ (defined in Eq. 12) for steps $t - 2$ and $t - 1$ (previous). Results highlight sample-dependent dynamic redundancy, with $V_{\text{ASR}}$ reflecting the degree of redundancy. (Redline marks the subsequent step where block skipping can begin.)

semantic importance from the spatial attention block, which guides token pruning across subsequent camera and motion attention blocks within the same denoising step. Then, we pick the values from the block-level rolling cache to fill the corresponding pruned attentions.

*At the chain level,* based on our observation that the denoising features gradually become stable at the later steps in the diffusion process, and the inter-step similarity can indicate the chain-level redundancy, we propose an *adaptive SCM chain bypass scheduler* to dynamically skip the entire SCM attention chain. Specifically, when the average similarity rate of the rolling cache meets the threshold, we schedule the denoising features to skip intermediate SCM chains and pass only through the first and last SCM chains, thereby significantly mitigating 4D generation latency.

### 4.2. Block Attention Reuse with Rolling Cache

As discussed in Challenge 1, Section 3, the SCM attention outputs across adjacent steps exhibit high cosine similarity, particularly during the later denoising steps. Motivated by this observation, we design a *rolling cache* mechanism to store the outputs of spatial, camera, and motion attention blocks for reusing in the subsequent denoising step. Moreover, each UNet-like layer is equipped with a rolling cache with its own shape, eliminating the need to track the sequencing index and substantially accelerating generation while enabling easy maintenance.

The rolling cache $\mathbf{\Omega}$ follows a first-in-first-out (FIFO) policy, in which cached features are inserted and popped in the order they were inserted. Specifically, during a dense denoising step $t - 1$ (where the proposed acceleration is not applied), the outputs of the SCM attention blocks at the current layer are sequentially stored into the cache as follows:

$$\mathbf{\Omega} \leftarrow \{\boldsymbol{A}_s^{t-1}, \boldsymbol{A}_c^{t-1}, \boldsymbol{A}_m^{t-1}\}, \tag{5}$$

where $\boldsymbol{A}_s^{t-1}$, $\boldsymbol{A}_c^{t-1}$, and $\boldsymbol{A}_m^{t-1}$ are the attention outputs of the spatial $\mathcal{F}_s$, camera $\mathcal{F}_c$, and motion $\mathcal{F}_m$ blocks, respectively. In the subsequent denoising step $t$, we retrieve the rolling cache $\mathbf{\Omega}$ with no additional computation costs

needed:

$$\{\tilde{\boldsymbol{A}}_s^t, \tilde{\boldsymbol{A}}_c^t, \tilde{\boldsymbol{A}}_m^t\} \leftarrow \mathbf{\Omega}. \tag{6}$$

Therefore, the forward computation in Eq. 3 within the block $\mathcal{F}$ at step $t$ can be reformulated as:

$$\mathcal{F}^t(\boldsymbol{Z}^t) = \text{FFN}(\boldsymbol{Z}^t + \tilde{\boldsymbol{A}}^t). \tag{7}$$

Notably, cached features are released sequentially to free memory, which can significantly reduce memory usage.

### 4.3. Semantic-Aware and Cache-Enhanced Pruning

Challenge 2 reveals that performing token pruning independently in each SCM attention block can disrupt the semantic-spatiotemporal coherence in 4D generation. To address this, we propose a *semantic-aware token identification* approach to select essential tokens in the entire SCM chain, and a *cache-enhanced pruning and refilling* strategy is then designed to address the considerable computational workload of camera and motion blocks while maintaining the spatiotemporal consistency, as shown in Figure 8 (see **Appendix C** for further discussion).

**Semantic-Aware Token Identifying.** Given the spatial cross-attention map $\boldsymbol{Q}_s \in \mathbb{R}^{F \times V \times H \times W}$ generated by the spatial attention block $\mathcal{F}_s$ at denosing step $t$, this map represents the semantic importance of each token in the generated multi-view video. Therefore, we identify the essential tokens for each image by applying the top-$K$ selection:

$$\boldsymbol{I}_c[f, :, :] = \text{argtop}_K\Big(\frac{1}{V}\sum_{v=1}^{V}\boldsymbol{Q}_s[f, v, :, :]\Big), \tag{8}$$

$$\boldsymbol{I}_m[v, :, :] = \text{argtop}_K\Big(\frac{1}{F}\sum_{f=1}^{F}\boldsymbol{Q}_s[f, v, :, :]\Big). \tag{9}$$

Here, $\boldsymbol{I}_c \in \mathbb{R}^{F \times H' \times W'}$ and $\boldsymbol{I}_m \in \mathbb{R}^{V \times H' \times W'}$ denote the indices of the selected tokens for camera and motion atten-

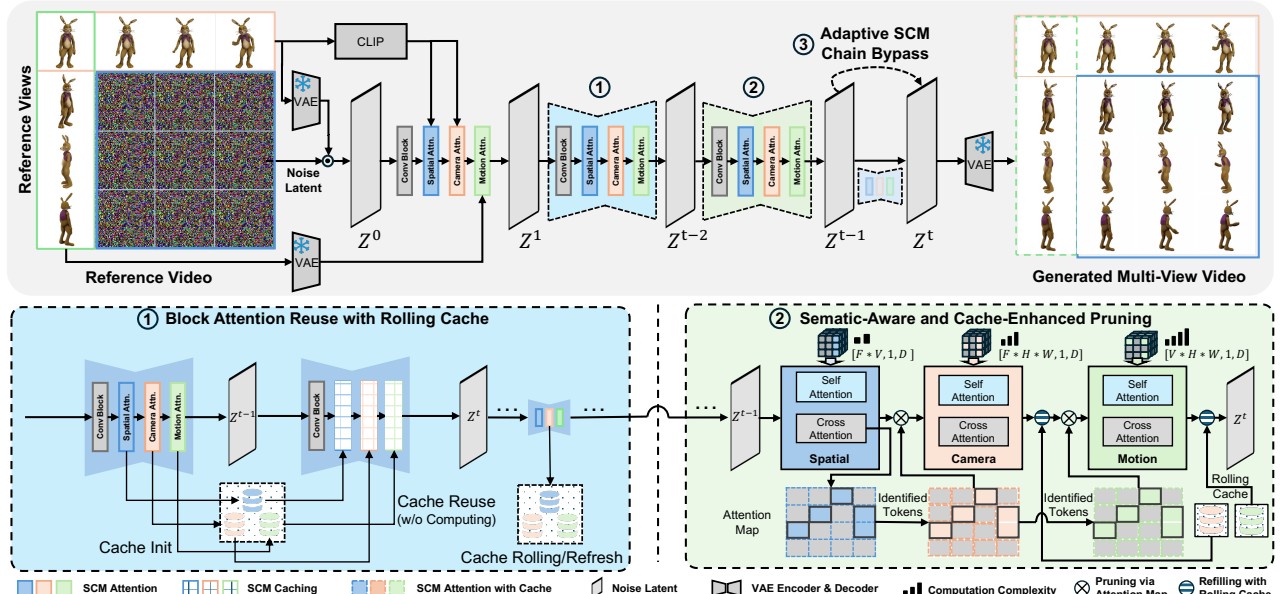

*Figure 8.* Overview of Turbo4DGen. During the 4D generation process, Turbo4DGen adopts a multi-level acceleration scheme across denoising steps. Specifically, the attention-level computations for the noise latent $Z^{t-1}$ are skipped by reusing attention outputs from the rolling cache. Then, we identify token-level redundancy to prune the computations for the camera and motion blocks in the next timestamp. Furthermore, as the denoising process proceeds, the entire SCM chain is adaptively bypassed for further acceleration.

tion blocks, respectively, and $K := H' \times W'$ represent the number of top tokens selected.

**Cache-Enhanced Pruning and Refilling.** Guided by the semantic importance, we can sparsify the camera and motion attention. Moreover, since each attention block requires reshaping back to its original spatial dimensions, we design a cache-enhanced refilling scheme on the pruned attentions, ensuring generation quality (as discussed in Section 3). The pruned attentions are replaced with rolling cache instead of fixed padding values (e.g., zero padding) as shown in Figure 8. We formulate this as:

$$\boldsymbol{A}_c^t = [\underbrace{\text{Attn}(\boldsymbol{Z}_s^t[\boldsymbol{I}_c], \boldsymbol{K}_c[\boldsymbol{I}_c])}_{\text{Compute}}, \underbrace{\tilde{\boldsymbol{A}}_c^t[\overline{\boldsymbol{I}}_c]}_{\text{Cache}}], \quad (10)$$

$$\boldsymbol{A}_m^t = [\underbrace{\text{Attn}(\boldsymbol{Z}_c^t[\boldsymbol{I}_m], \boldsymbol{K}_m[\boldsymbol{I}_m])}_{\text{Compute}}, \underbrace{\tilde{\boldsymbol{A}}_m^t[\overline{\boldsymbol{I}}_m]}_{\text{Cache}}]. \quad (11)$$

Here, $\tilde{\boldsymbol{A}}_c^t$ and $\tilde{\boldsymbol{A}}_m^t$ represent the cached camera and motion attention outputs from the previous step. $\overline{\boldsymbol{I}}_c$ and $\overline{\boldsymbol{I}}_m$ are the indices of the pruned tokens retrieved from the rolling cache. The overall algorithm is provided with Python-style pseudo-code (See **Appendix D** for further discussion).

### 4.4. Adaptive SCM Chain Bypass Scheduling

To reduce chain-level redundancy, we propose an adaptive SCM chain attention bypassing scheduler based on the inter-step cache similarity, as discussed in Section 3. Turbo4DGen utilizes the similarity between two-step

rolling cache $\boldsymbol{\Omega}$ as an indicator for the chain-level redundancy degree. Specifically, we calculate the *average similarity rate* $V_{\text{ASR}}$ of rolling cache $\boldsymbol{\Omega}$ between two denoising steps (refer to the scheduling policy in **Appendix E**)

$$V_{\text{ASR}} = \sum_{i \in \{s,c,m\}} \sum_{j=t-\Delta t}^{t} \cos(\tilde{\boldsymbol{A}}_i^j, \boldsymbol{A}_i^j), \quad (12)$$

where $\Delta t$ represents the number of steps counted backward from the current step $t$, and $\cos(\cdot, \cdot)$ computes the cosine similarity. When the $V_{\text{ASR}}$ exceeds the threshold $\alpha$, it represents the high stability of denoising features. Then, the scheduler enables denoising features into the first and last paired chains, bypassing the intermediate chains. Exploiting a scheduler rather than a fixed hyperparameter dynamically reduces redundant computation in subsequent steps while preventing unwarranted skipping of important chains.

## 5. Experiments

**Implementation Details.** We implement the proposed Turbo4DGen based on the SV4D (Xie et al., 2025) codebase. We adopt the SV3D$_p$ (Voleti et al., 2024) checkpoint as the multi-view generation model and the SV4D checkpoint as the baseline for 4D generation. We conduct the experiments on our servers equipped with 8 NVIDIA RTX PRO 6000 GPUs, 2 AMD EPYC 9254 CPUs (96 Cores), and 1.5 TB of RAM.

**Datasets.** We evaluate Turbo4DGen on the widely used ObjaverseDy (Deitke et al., 2022) and Consistent4D (Jiang

*Table 1.* Quantitatively experimental results on the ObjaverseDy (Deitke et al., 2022) dataset, in comparison with the baseline, SV4D (Xie et al., 2025), and other stat-of-the-art methods. The second-best results are underlined.

| Method | LPIPS↓ | CLIP-S↑ | PSNR↑ | SSIM↑ | FVD-F↓ | FVD-V↓ | FVD-Diag↓ | FV4D↓ |
|--------|--------|---------|-------|-------|--------|--------|-----------|-------|
| Consistent4D (Jiang et al., 2023) | 0.148 | 0.899 | 16.44 | 0.866 | 781.38 | 510.04 | 782.79 | 658.31 |
| STAG4D (Zeng et al., 2024) | 0.155 | 0.868 | 16.73 | 0.867 | 848.83 | 539.96 | 709.52 | 833.08 |
| DG4D (Ren et al., 2023) | 0.156 | 0.874 | 16.02 | 0.860 | 826.72 | 543.29 | 761.58 | 741.99 |
| L4GM (Ren et al., 2024) | 0.146 | 0.902 | 17.65 | 0.877 | 805.87 | 537.46 | 782.92 | 666.48 |
| SV4D (Xie et al., 2025) | 0.122 | 0.902 | 18.47 | 0.884 | 754.23 | 436.86 | 666.59 | 699.04 |
| Turbo4DGen (**9.7× Speedup**) | **0.113** | **0.917** | **20.27** | **0.891** | **626.36** | **410.93** | **642.26** | **549.03** |

*Table 2.* Quantitative results on the Consistent4D (Jiang et al., 2024) dataset, in comparison with the baseline, SV4D (Xie et al., 2025), and other state-of-the-art methods. Note that the results of Turbo4DGen is **zero-shot** on this dataset.

| Method | LPIPS↓ | CLIP-S↑ | FVD-F↓ |
|--------|--------|---------|--------|
| Consistent4D (Jiang et al., 2023) | 0.160 | 0.87 | 1133.93 |
| STAG4D (Zeng et al., 2024) | 0.126 | 0.91 | 992.21 |
| 4Diffusion (Zhang et al., 2024) | 0.165 | 0.88 | – |
| Efficient4D (Pan et al., 2024) | 0.130 | 0.92 | – |
| L4GM (Ren et al., 2024) | 0.120 | 0.94 | 691.87 |
| SV4D (Xie et al., 2025) | **0.118** | 0.92 | 732.40 |
| Turbo4DGen (**9.7× Speedup**) | 0.119 | 0.93 | 708.51 |

et al., 2023) datasets. In particular, the Consistent4D (Jiang et al., 2023) dataset contains dynamic objects collected from ObjaverseDy (Deitke et al., 2022), which we exclude from our training set to ensure a fair comparison. Additionally, we apply the filtering algorithm proposed by Diffusion4D (Liang et al., 2024a) to remove the static 3D objects for our experiments.

**Fine-Tuning.** For the ObjaverseDy (Deitke et al., 2022) dataset, we fine-tune Turbo4DGen on a subset of data using the loss function (Xie et al., 2025), $\mathbb{E}_{\sigma,y,n} \left[ \lambda(\sigma) \| \mathcal{D}(y + n; \sigma) - y \|_2^2 \right]$, where $\sigma$ denotes noise level, $y$ represents the training data, and $n \sim \mathcal{N}(\mathbf{0}, \sigma^2\mathbf{I})$ is Gaussian noise. The tuning set contains over 11,000 filtered 4D objects, each represented by 40 multi-view images with data preprocessing, i.e., extracting multi-view images from 3D models, generating CLIP (Radford et al., 2021) embeddings, and computing VAE (Cemgil et al., 2020) latent, etc. For the Consistent4D (Jiang et al., 2024) dataset, we conduct **zero-shot** evaluation without any tuning.

**Generation Quality Metrics.** We generate novel-view videos along the trajectories of ground-truth cameras in the evaluation datasets and compare each frame with its corresponding ground-truth frame. Following the baseline, we use standard metrics: CLIP-Score (CLIP-S) (Radford et al., 2021) for visual quality, Learned Perceptual Similarity (LPIPS) (Zhang et al., 2018) for perceptual similarity (lower is better), Peak Signal-to-Noise Ratio (PSNR) (Horé

& Ziou, 2010) and SSIM (Wang et al., 2004) for pixel-level fidelity (higher is better), and FVD (Unterthiner et al., 2019) for video coherence. FVD is computed in multiple ways: FVD-F over frames at each view, FVD-V over views at each frame, FVD-Diag over diagonal images, and FV4D over all images in a bidirectional raster scan.

**Efficiency Metrics.** We measure the total latency for each 4D generation given a monocular video input, report the average across all samples, and compute the corresponding speedups against the baseline. We further evaluate peak memory usage during generation and report the average.

**Hyperparameter Settings.** We set the generation resolution as 576×576, and set the frames and views as 5 and 8, respectively. The number of denoising steps and elevation angle are set as 20 and $30°$, respectively. The top-$K$ ratio is set to 0.2 (selecting 20% of tokens along the $H$ and $W$ axes), the ASR trace step number $\Delta t$ to 3, and the bypassing threshold $\alpha$ to 0.9, while other parameters remain at their default settings.

### 5.1. Main Results

**Efficiency.** We report the latency speedup and peak memory usage of Turbo4DGen compared to the baseline SV4D (Xie et al., 2025) and other state-of-the-art methods on the ObjaverseDy (Deitke et al., 2022) dataset, across various generated video sizes, under the same denoising step settings, as shown in Figure 9. Note that DeepCache (Ma et al., 2024) and AT-EDM (Wang et al., 2024b) are originally designed for 2D or video generation and are not directly applicable to 4D tasks. For fair comparison, we extend their implementations to the 4D generation setting. As shown in our experiments, both the baseline and DeepCache (Ma et al., 2024) encounter out-of-memory (OOM) failures when generating 4D scenes with 30 frames and 8 views. Although DeepCache (Ma et al., 2024) reduces latency through block-based caching, its lack of pruning leads to higher peak memory usage. In contrast, AT-EDM (Wang et al., 2024b) effectively reduces memory via single-attention pruning but suffers from substantial performance degradation. In comparison, our Turbo4DGen achieves an average speedup of 9.7× while significantly reducing memory consumption and

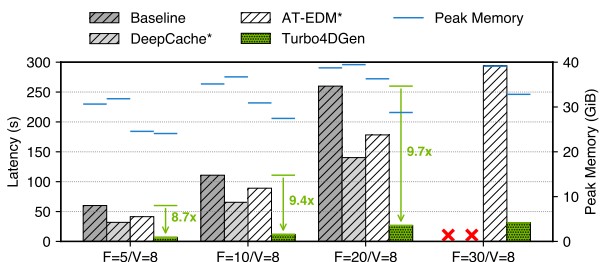

*Figure 9.* The latency, speedup, and peak memory evaluation are presented, where "**X**" indicates OOM errors, and $F$ and $V$ denote the number of generated frames and views, respectively. Methods marked with "*" are not directly applicable to 4D generation, and their code is modified for experimental evaluation.

preserving generation quality.

**Quantitative Quality.** As summarized in Table 1 and Table 2, we quantitatively evaluate the generation quality using the standard metrics (e.g., LPIPS, CLIP-S, PSNR, SSIM, FVD) and compare Turbo4DGen with existing state-of-the-art approaches on the ObjaverseDy (Deitke et al., 2022) and Consistent4D (Jiang et al., 2024) datasets. Turbo4DGen can achieve state-of-the-art (SOTA) performance across all metrics while maintaining high generation quality (see the visual comparison in the next section). However, the Consistent4D (Jiang et al., 2024) dataset provides ground truth for a tiny subset of test data, while the training data contains no ground truth and consists solely of input videos. Therefore, we perform **zero-shot** evaluation. The results indicate that our method still achieves high performance (slightly behind the best) on this challenge dataset.

**Visual Quality.** We present visual quality comparisons by displaying five frames from the input videos and their corresponding novel views in **Appendix** G.1. Compared to the baseline method, SV4D Turbo4DGen preserves geometric structures and texture details, generating synthesized views with strong spatiotemporal consistency across frames for both simple (top) and complex (bottom) structured 4D objects. Specifically, for the sample structure object, Turbo4DGen maintains fine details, e.g., shape, shadow, and eye regions, without sacrificing generation quality. For the complex case, Turbo4DGen preserves fine-textured regions and ensures coherent motion consistency without noticeable blur or details lost. Overall, while minor visual imperfections remain in some cases, Turbo4DGen substantially reduces generation latency, achieving an effective balance between efficiency and visual fidelity.

### 5.2. Ablation Study

**Ablation of Different Strategies.** We evaluate the effects of pruning, caching and bypass strategies in Turbo4DGen and present the results in **Appendix** F.1. We observe that the cache-enhanced pruning mechanism effectively preserves

generation quality and saves memory, although pruning still requires computing the object semantic tokens, resulting in a slight speedup. In contrast, the SCM caching and bypass can substantially accelerate the process, even though it skips the computation of some essential tokens and, as a result, degrades generation quality. This highlights the trade-off between efficiency and performance. We further analyze the sensitivity to warm-up steps in **Appendix** F.2.

**Ablation of Different Pruning Rates.** We analyze the influence of varying pruning rates in the proposed token pruning mechanism. As shown in **Appendix** F.3, moderate pruning achieves an optimal trade-off between efficiency and generation quality, while a higher pruning rate leads to noticeable degradation in generation quality and temporal consistency (as measured by PSNR). In particular, an aggressive pruning rate results in blurrier and less smooth generation results.

**Ablation of Different Chain Bypassing Thresholds.** We evaluate the influence of varying chain bypassing thresholds ranging from 0.8 to 0.95 (as discussed in Section 3), while keeping all other parameters at their default settings. From **Appendix** F.4, it is seen that lower thresholds trigger earlier chain bypassing, resulting in higher speedups but causing some essential steps to be skipped, leading to a degradation in generation quality. In contrast, larger thresholds yield limited additional speedup while maintaining similar generation quality.

**Semantic-Aware Pruning vs. Random Pruning** We study the effectiveness of the proposed semantic-aware token pruning vs. random pruning. Random token pruning leads to a substantial degradation in generation quality. In contrast, our semantic-aware pruning method, combined with cache reuse, effectively preserves generation quality, as reported in **Appendix** F.5.

## 6. Conclusion

In this paper, we propose Turbo4DGen, an ultra-fast acceleration framework for 4D generation. Turbo4DGen identifies and removes redundant computations at multi-scale granularity in the SCM attention mechanism – i.e., token, block, and chain levels – with the designed rolling cache and adaptive bypassing mechanism, while maintaining high generation quality. To the best of our knowledge, this is the first systematic framework for accelerating 4D generation. Our method achieves an average speedup of 9.7× over the baseline, demonstrating the superiority of Turbo4DGen over state-of-the-art approaches.

## Acknowledgment

This work was partially supported by the National Science Foundation under Award 2343618.

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

## A. Impact Statement

Turbo4DGen aims to make 4D generation more accessible by reducing the compute and memory needed for dynamic 3D content synthesis. Because the framework is general and can be adapted to DiT-based generation methods, we hope it will be useful beyond a single system. More efficient generation may help researchers, educators, and creators experiment with 4D content using more accessible hardware, while also lowering the energy cost of repeated inference.

## B. Related Works

**Multi-View Generation.** Multi-view generation focuses on synthesizing novel views of a scene from limited observations, and has evolved from geometry-based methods to deep learning approaches. Models such as Neural Radiance Fields (NeRF) (Mildenhall et al., 2020) and its variants achieve high-fidelity results but often require dense input and are computationally expensive. More recent methods, such as SV3D (Voleti et al., 2024), leverage 3D-aware representations within diffusion models to improve view consistency and generation quality, demonstrating strong performance in static multi-view generation tasks. Extensions to dynamic scenes, such as NeVRF (Xian et al., 2021), MoVideo (Liang et al., 2024b) and DreamVideo (Wang et al., 2024a), introduce temporal modeling to ensure frame coherence but face challenges in scalability and efficiency. Additionally, multi-modal models like CogVideo (Hong et al., 2022), Make-A-Video (Singer et al., 2022), Video LDM (Blattmann et al., 2023b), Latent-Shift (An et al., 2023), and VideoDiffusion (Luo et al., 2023) incorporate text or depth inputs, further increasing complexity. Compared with the multi-view generation task, the 4D generation task focused on in our paper is more challenging due to the additional temporal dimension, especially when generating longer videos.

**4D Generation.** 4D generation, or dynamic 3D content generation, has gained significant attention for its ability to synthesize temporally coherent 3D scenes across multiple views and time steps. Recent academic efforts such as SV4D (Xie et al., 2025) and CAT4D (Wu et al., 2025) extend diffusion models to the dynamic multi-view setting by incorporating spatiotemporal and view-consistent attention mechanisms, enabling coherent frame synthesis across time and viewpoints. Similarly, 4Real (Yu et al., 2024a) advances dynamic 3D generation by integrating real-world priors and efficient rendering strategies. While these methods represent important steps toward high-quality 4D synthesis, they remain computationally intensive due to dense attention over spatial, temporal, and view dimensions. In parallel, several industry models have demonstrated impressive 4D generation capabilities; however, most of these systems are not publicly released, limiting reproducibility and broader research progress. In contrast, our work addresses key computational bottlenecks by reducing redundancy in SCM attention computations via a designed rolling cache and an adaptive bypassing mechanism, thereby making 4D generation more efficient and accessible.

**Diffusion Model Acceleration.** Despite the impressive generative capabilities, diffusion models are computationally expensive due to their iterative denoising process. To address this, recent research has increasingly focused on accelerating inference in diffusion models. Knowledge distillation (KD)-based approaches, such as BK-SDM (Kim et al., 2024), DKDM (Xiang et al., 2025), and DiffKD (Huang et al., 2023), distill the multi-step diffusion process into a smaller student model, enabling faster sampling with fewer steps. These methods achieve good speed-quality trade-offs but require costly retraining and are sensitive to the distillation objective. In contrast, DeepCache (Ma et al., 2024) proposes reusing intermediate features across consecutive denoising steps to exploit temporal redundancy. AT-EDM (Wang et al., 2024b) is an attention-aware method that prunes tokens within a single denoising step, further enhancing the efficiency of diffusion models. Nonetheless, these approaches are only applicable to simple UNet-like architectures and fail to generalize effectively to more complex 4D generation scenarios. Our work complements this line of research by targeting 4D generation and addressing attention redundancy when generating multi-view video at multi-scale attention granularity.

## C. Additional Discussion on Cache-Enhanced Semantic-Aware Pruning

Independent token pruning in each SCM attention block, e.g., AT-EDM* (Wang et al., 2024b), can break semantic alignment across views and frames, leading to inconsistent geometry or motion. To address this, we perform semantic-aware token selection across the SCM chain using spatial attention (motivated by our observation as shown in Figure 5), ensuring that all SCM blocks operate on a consistent set of essential tokens, which is crucial for 4D generation. To further reduce computation, attention outputs for pruned tokens in camera and motion blocks are retrieved from a rolling cache instead of being recomputed. This cache-enhanced design preserves semantic-spatial-temporal consistency while significantly accelerating inference, achieving a good balance between efficiency and visual quality (as shown in Figure 10).

As described in Eq. 4, the SCM computing chain is formulated as

$$\boldsymbol{A}^t = \mathcal{F}_m^t\Big(\mathcal{F}_c^t\big(\mathcal{F}_s^t(\boldsymbol{Z}^t,\ \boldsymbol{K}_s),\boldsymbol{K}_c\big),\boldsymbol{K}_m\Big),$$

where $\boldsymbol{K}_s < \boldsymbol{K}_c < \boldsymbol{K}_m$ (the size). After pruning (as discussed in Section 5), we select the top-$K$ tokens along the $H$ and $W$ axes for $\boldsymbol{Z}^t$, $\boldsymbol{K}_c$, and $\boldsymbol{K}_m$ (i.e., vae$_{\text{factor}} = 8$, $576 \to 72$, $F = 5$, $V = 8$, $d = 1024$, $b = 2$ for CFG conditional and unconditional tokens). As a result, $\boldsymbol{Z}^t$ is reduced from $[2{\times}5{\times}8, 72, 72, 1024]$ to $[2{\times}5{\times}8, 14, 14, 1024]$, the camera context shrinks from $[2{\times}5, 72, 72, 1024]$ to $[2{\times}5, 14, 14, 1024]$, and the motion context from $[2{\times}8, 72, 72, 1024]$ to $[2{\times}8, 14, 14, 1024]$ (reshape omitted for clarity). This reduces camera and motion attention FLOPs by $\approx 96.2\%$ each. A similar reduction occurs for the cache, while the rolling cache is retained, it decreases peak memory by $\approx 24.2\%$ (measured via the PyTorch API).

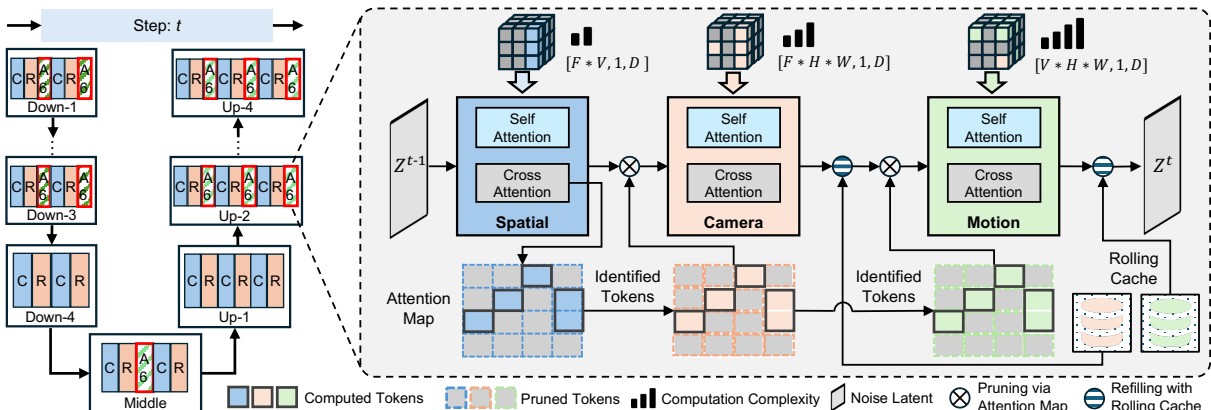

*Figure 10.* Illustration of the cache-enhanced semantic-aware pruning mechanism. With the noise latent $\boldsymbol{Z}^{t-1}$ as input of the denoising step $t$, the spatial attention block first generates the attention map representing the semantic importance of tokens, which is then adopted for the pruning of subsequent camera and motion blocks with intensive computation complexity. Moreover, the attention outputs of the camera and motion blocks are refilled from the rolling cache, leading to faster generation speed while preserving semantic-spatiotemporal consistency.

## D. Pseudocode of Semantic-Aware Cache-Enhanced Pruning Algorithm

We provide Python-style pseudocode of the semantic-aware cache-enhanced pruning algorithm. For simplicity, we show a sample implementation focusing on the spatial attention token identification.

**Algorithm 1** Semantic-Aware and Cache-Enhanced Pruning

```
1  def pruning(Q_s, Z_c, K_c, K_m, cache):
2      # Semantic-aware token identifying
3      I_c, I_m = identifying(Q_s, topk)
4      # Cache-enhanced refilling
5      refill_c, refill_m = cache.get()
6      # Forward process
7      Z_c, K_c = get_element(Z_c, K_c, I_c)
8      A_c = [attn_op(Z_c, K_c), refill_c]
9      Z_m, K_m = get_element(Z_m, K_m, I_m)
10     A_m = [attn_op(Z_m, K_m), refill_m]
11     return A_m
12
13 def identifying(Q_s, topk):
14     # Input: Spatial map Q_s, number topk
15     # Output: Token indices I_c, I_m
16     _, I_c = torch.topk(Q_s.mean(1), k=topk, dim=-1)
17     _, I_m = torch.topk(Q_s.mean(0), k=topk, dim=-1)
18     return I_c, I_m
```

## E. Scheduling Policy for Cache/Pruning/Bypass

As shown in Figure 8, we start with **three fully-computed warm-up steps** to refresh the rolling cache. We then perform interleaved **block**-cache steps (reusing cache without attention computation) and pruning steps (refilling uncomputed **tokens** from cache and refreshing cache). At each step, we compute $V_{ASR}$ (eq. 12), once it exceeds threshold $\alpha$, subsequent steps switch to bypass mode, where only the first and last UNet blocks are computed and intermediate blocks are skipped (**chain**).

## F. Ablation Study

### F.1. Ablation of Different Strategies

*Table 3.* Ablation study on different efficiency granularities for acceleration, reporting results in terms of pruning, caching, and bypass. The bold number indicates the best performance.

| Method | Speedup↑ | LPIPS↓ | PSNR↑ |
|---|---|---|---|
| SV4D (Baseline) | 1.00× | 0.122 | 18.47 |
| Turbo4DGen w/ pruning | 2.28× | **0.106** | **22.86** |
| Turbo4DGen w/ caching | 5.25× | 0.121 | 19.48 |
| Turbo4DGen w/ bypass | **11.80×** | 0.149 | 18.75 |
| Turbo4DGen | 9.70× | 0.113 | 20.27 |

### F.2. Ablation of Warm-up Steps

To evaluate the sensitivity of our approach to warm-up steps, we tested starting from different steps (denoising step = 20, same as SV4D) using the fine-tuned checkpoint. As shown in Table 4, the performance remains largely unaffected, indicating that the model is insensitive to the choice of warm-up step.

*Table 4.* Ablation study of warm-up steps, with results reported in efficiency and generation quality. The yellow color indicates our default setting.

| Warm-up Steps | Speedup↑ | LPIPS↓ | PSNR↑ | CLIP-S↑ | SSIM↑ |
|---|---|---|---|---|---|
| **3** | **9.70×** | **0.113** | **20.27** | **0.917** | **0.891** |
| 2 | 9.75× | 0.115 | 19.86 | 0.909 | 0.885 |
| 1 | 9.81× | 0.118 | 19.13 | 0.904 | 0.883 |

### F.3. Ablation of Different Pruning Rates

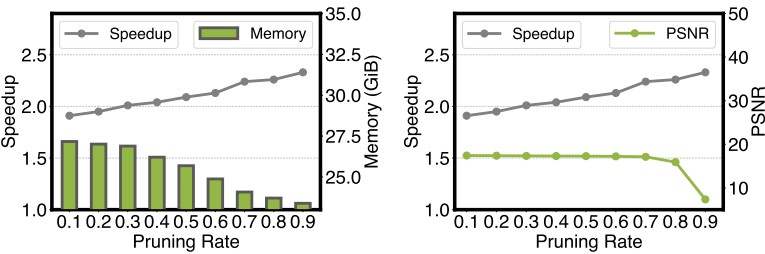

*Figure 11.* Influence of varying pruning rates, with results reported in terms of speedup, memory usage, and PSNR.

### F.4. Ablation of Different Chain Bypass Thresholds

*Table 5.* Ablation study on chain bypassing thresholds, with results reported in efficiency and generation quality. The yellow color indicates our default setting.

| Threshold $\alpha$ | Speedup↑ | PSNR↑ | CLIP-S↑ | SSIM↑ |
| --- | --- | --- | --- | --- |
| 0.80 | 12.11 | 15.31 | 0.838 | 0.824 |
| 0.85 | 11.73 | 17.81 | 0.863 | 0.855 |
| **0.90** | **9.70** | **20.27** | **0.917** | **0.891** |
| 0.95 | 9.56 | 20.29 | 0.919 | 0.893 |

## F.5. Semantic-Aware Pruning vs. Random Pruning

*Table 6.* Ablation study of different pruning strategies, with results reported in terms of LPIPS, CLIP-S, PSNR, and SSIM.

| Method | LPIPS↓ | CLIP-S↑ | PSNR↑ | SSIM↑ |
| --- | --- | --- | --- | --- |
| Random | 0.188 | 0.563 | 10.85 | 0.571 |
| Turbo4DGen | **0.130** | **0.828** | **17.13** | **0.849** |

# G. Additional Results

## G.1. Visual Quality

In Figure 12 and Figure 13, we present additional 4D generation results on the Objaverse (Deitke et al., 2022) dataset. The top row corresponds to a simple object, while the bottom row illustrates a complex one. For each case, we sample five input frames and their corresponding novel views for comparison among the baseline method, SV4D (Xie et al., 2025), and Turbo4DGen. The results demonstrate that Turbo4DGen achieves high-fidelity generation with consistent geometry and texture details across views.

# H. Limitations

Our framework is optimized for standard 4D generation settings ($\geq 20$ denoising steps) and SCM-based architectures, where it demonstrates robust performance. However, its effectiveness may degrade in very low-step settings or in scenarios with rapid object motion (e.g., rotation), where reduction $V_{ASR}$ makes it harder to trigger the bypass threshold and maintain efficiency-consistency trade-offs.

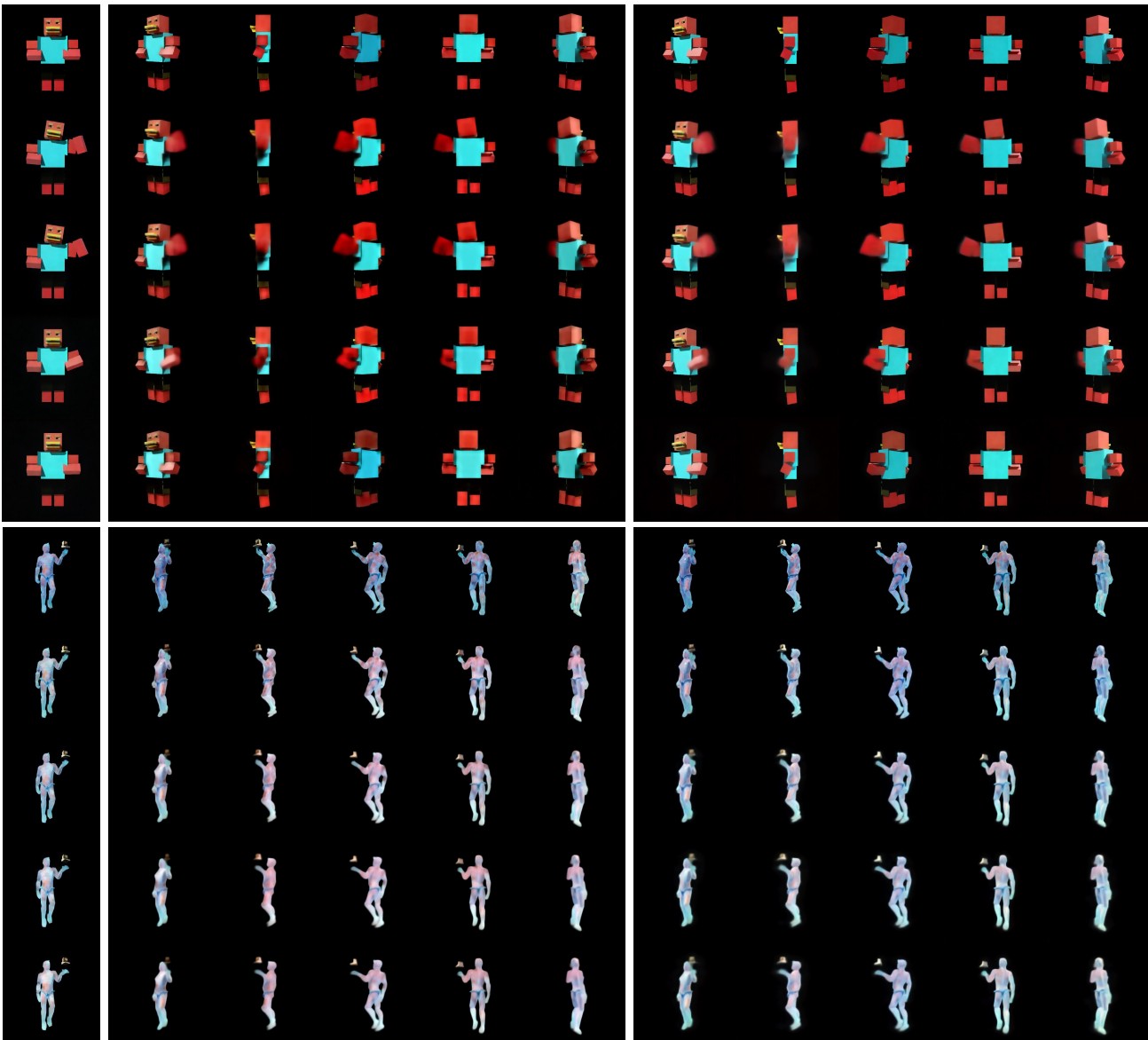

*Figure 12.* Visual quality comparison. (**Left**) Monocular input video; (**Middle**) Results of SV4D (Xie et al., 2025); (**Right**) Result of Turbo4DGen.

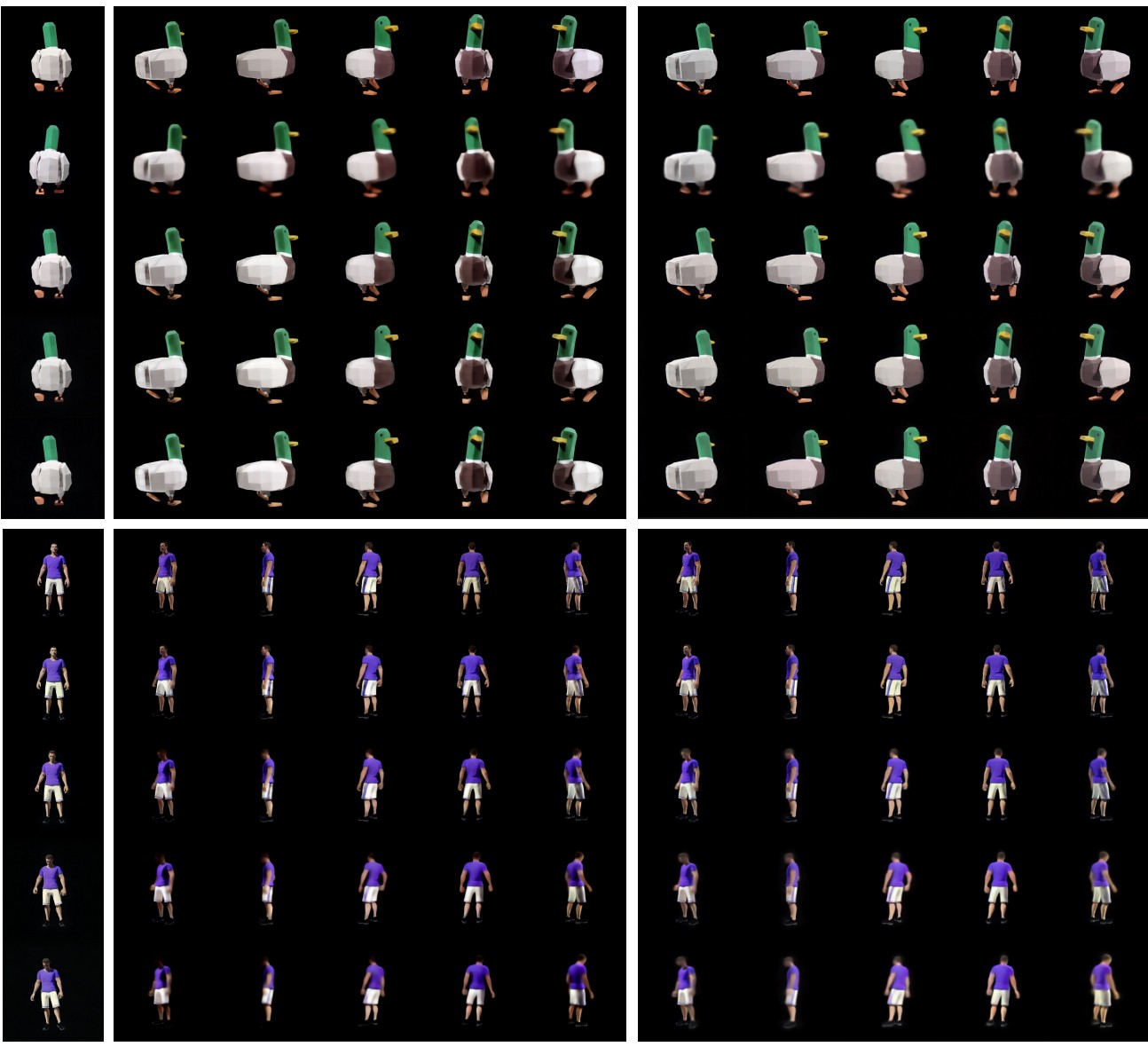

*Figure 13.* Visual quality comparison. (**Left**) Monocular input video; (**Middle**) Results of SV4D (Xie et al., 2025); (**Right**) Result of Turbo4DGen.

