# OpenReview forum: "Turbo4DGen: Ultra-Fast Acceleration for 4D Generation"
_ICML.cc/2026/Conference — ICML 2026 regular_

### Official Review · Reviewer_9JhK · 2026-03-06

**Soundness:** 3
**Presentation:** 2
**Significance:** 3
**Originality:** 2
**Overall Recommendation:** 4
**Confidence:** 3

**Summary:**

The paper proposes an acceleration framework for 4D content generation models that employ Spatial-Camera-Motion attention chains. Combined with dynamically semantic-aware attention pruning and an adaptive SCM chain bypass scheduler, this framework achieves accelerated 4D object generation.

**Compliance With Llm Reviewing Policy:**

Affirmed.

**Final Justification:**

My main concern is only partially resolved, since the evidence for generalization beyond the original SV4D setting remains relatively limited and much of the new support is currently confined to the rebuttal rather than the paper itself. Overall, I view this as a promising paper that is closer to borderline accept after rebuttal.

**Key Questions For Authors:**

Finally, since this is a 4D generation task, without accompanying videos/visual results, it is difficult to evaluate the actual visual quality of the model and the specific impact of the ablation studies.

**Limitations:**

No, the authors have not adequately discussed the limitations and potential negative societal impacts (see **Section 6**).

 I believe the method is still too limited to a single network architecture, and there is no discussion regarding generalization experiments.

**Strengths And Weaknesses:**

**Strengths:**

1. The authors address an important efficiency problem in 4D generation tasks.

2. The proposed block-level rolling cache, semantic-aware token selection, and chain-level bypassing make sense to me.

**Weaknesses**:

1. My biggest concern is that the experimental comparisons in this work are unfair. Specifically, Turbo4DGen is implemented based on SV4D and undergoes additional fine-tuning on ObjaverseDy. This leads to unfair comparisons in the results shown in Table 1 and Table

2. The paper only demonstrates results based on a single base model (e.g., SV4D). It does not explore different backbones or more generalized scenarios with more complex inputs (in-the-wild datasets). Potentially missing citations: Free4D and Diff4Splat. Demonstrating the framework's effectiveness in scene-level generation would help strengthen the paper's advantages.

[1] Diff4Splat: Controllable 4D Scene Generation with Latent Dynamic Reconstruction Models

[2] Free4D: Tuning-free 4D Scene Generation with Spatial-Temporal Consistency

3. The Average Similarity Rate ($V_{\text{ASR}}$) is only evaluated at a specific value. I think more experiments are needed to fully demonstrate its effectiveness. And I think it is important for this paper.

---

> ### Author Rebuttal · Authors · 2026-03-31
>
> We sincerely thank you for the thoughtful feedback.
>
> ###  Fine-Tuning
>
> Thanks for pointing it out. Our evaluations in Section 5 consist of both fine-tuning and no fine-tuning. Table 1 shows the evaluation on ObjaverseDy with fine-tuning, while Table 2 shows the evaluation on Consistent4D **without fine-tuning**. Consistent4D is a widely used benchmark for 4D generation research that releases only the test set ground truth, making it a very appropriate testbed for evaluating the zero-shot ability of our method.
>
> We conducted additional experiments on ObjaverseDy without fine-tuning in Table R5, which shows Turbo4DGen still brings significant speedup.
>
> **Table R5**. Eval. w/o tuning.
> |Method|Speedup|LPIPS|PSNR|CLIP-S|SSIM|
> |---|---|---|---|---|---|
> |SV4D|1.0|0.122|18.47|0.902|0.884|
> |Turbo4DGen(Ours)|5.8|0.125|18.36|0.891|0.881|
>
> ###  Baseline Selection&More Dataset
>
> We appreciate you bringing the related works, i.e., DiffSplat and Free4D, to us. We will introduce and discuss them in the revised manuscript. Our Turbo4DGen works on a different stage from those works: those works focus on 4D scene reconstruction based on generated 4D multi-views, while our work focuses on multi-view 4D generation. We cannot apply our work to those models.
>
> We have to select SV4D as our baseline because, to the best of our knowledge, it was the **only** state-of-the-art open-source 4D generation model. Other works like CAT4D and 4Real remain closed-source (model, dataset, etc.), which prevents us from building upon them.
>
> Our Turbo4DGen is a general optimization that accelerates the Spatio-Camera-Motion (SCM) attention mechanism in 4D generation, and our core multi-level computation skipping is designed as general, plug-and-play components. For other 4D generation models, e.g., CAT4D and 4Real, that use a similar attention mechanism, we believe Turbo4DGen can be easily applied to accelerate them.
>
> Due to rebuttal time limits, we did our best to apply Turbo4DGen to accelerate TrajectoryCrafter [R1], a notable model leveraging a similar SCM paradigm. The bottleneck is its SCM-like attention chain, which consists of a dual-stream conditional video diffusion process including Ref-DiT (cross-attention monocular input and camera-trajectory point clouds) and VideoDiT. Following the protocol in TrajectoryCrafter [R1], we evaluate our Turbo4DGen on the **in-the-wild** iPhone [R2] dataset **without fine-tuning**. As shown in Table R6, our method transfers effectively to alternative baselines while yielding stable performance. We will open-source our method and test more state-of-the-art models once they become public.
>
> **Table R6**. Eval. on **in-the-wild** iPhone[R2] dataset. Pruning rate=0.5.
> |Method|Scene|Speedup|PSNR(dB)|SSIM|LPIPS|
> |---|---|---|---|---|---|
> |TrajectoryCraft[R1]|Apple|1.00|13.88|0.285|0.612|
> ||Block|1.00|14.21|0.528|0.479|
> ||Paper|1.00|14.89|0.482|0.471|
> ||Spin|1.00|14.51|0.380|0.518|
> ||Teddy|1.00|13.73|0.411|0.513|
> ||*Mean*|*1.00*|*14.24*|*0.417*|*0.519*|
> |Turbo4DGen(Ours)|Apple|4.93|13.62|0.263|0.601|
> ||Block|5.14|13.96|0.506|0.467|
> ||Paper|4.70|14.65|0.461|0.458|
> ||Spin|5.20|14.24|0.359|0.507|
> ||Teddy|4.80|13.47|0.389|0.503|
> ||*Mean*|*4.95*|*14.00*|*0.396*|*0.507*|
>
> ### Average Similarity Rate
> Due to the page limit, we place the analysis of $V_{ASR}$ in Appendix E.4, where the thresholds for $V_{ASR}$ and the cache policy are set to be constant across all datasets. We conducted additional experiments to study the value, as shown in Table R7, which will be included in the revised version.
>
> **Table R7.**
> |Denois.Step|Resolution|Threshold|ObjaverseDy|||Consistent4D(w/oTuning)|||
> |---|---|---|---|---|---|---|---|---|
> ||||Speedup|PSNR|CLIP-S|Speedup|PSNR|CLIP-S|
> |20|576×576|0.85|11.73|17.81|0.863|11.54|17.09|0.889|
> |||*0.90*|*9.70*|*20.27*|*0.917*|*9.67*|*18.25*|*0.927*|
> |||0.95|9.56|20.29|0.919|9.42|18.28|0.928|
> ||768×768|0.85|11.98|17.46|0.866|11.63|16.92|0.893|
> |||*0.90*|*9.81*|*20.21*|*0.920*|*9.72*|*18.18*|*0.933*|
> |||0.95|9.63|20.25|0.924|9.47|18.19|0.935|
> |30|576×576|0.85|13.52|19.11|0.880|12.96|18.87|0.904|
> |||*0.90*|*10.63*|*21.08*|*0.922*|*10.41*|*19.07*|*0.929*|
> |||0.95|10.41|21.13|0.925|10.30|19.09|0.929|
> ||768×768|0.85|13.66|18.83|0.883|13.03|18.81|0.909|
> |||*0.90*|*10.72*|*21.01*|*0.924*|*10.51*|*19.02*|*0.930*|
> |||0.95|10.67|21.05|0.925|10.44|19.03|0.931|
>
> ### Visual Quality
> We have shown example frames to show the visual quality results in Appendix F, showing Turbo4DGen brings high-fidelity generation with consistent geometry and texture details across views.
> We further generated 4D videos using our Turbo4DGen on the iPhone dataset [R2]. Those example videos are at https://anonymous.4open.science/r/anon-code-x9j2/.
>
> ###  Limitations&Impact Statement
>
> We will add dedicated sections on them, referring to the corresponding responses to other reviewers.
>
> [R1] TrajectoryCrafter: Redirecting Camera Trajectory ..., ICCV 2025.
>
> [R2] Monocular Dynamic View Synthesis: A Reality Check, NeurIPS 2022.

---

> > ### Author Rebuttal · Reviewer_9JhK · 2026-04-03
> >
> > Thank you for the thorough rebuttal and the substantial additional experiments. I appreciate the clarifications on the fine-tuning protocol, the added
> >   zero-shot results, the transfer test on TrajectoryCrafter, and the threshold analysis for ($V_{\text{ASR}}$); these materially strengthen the paper and
> >   make the contribution more convincing. That said, my main concern is only partially resolved, since the evidence for generalization beyond the original
> >   SV4D setting remains relatively limited and much of the new support is currently confined to the rebuttal rather than the paper itself. Overall, I view
> >   this as a promising paper that is closer to borderline accept after rebuttal.
> >
> > so I raise my score.

---

> > > ### Author Response · Authors · 2026-04-05
> > >
> > > Thank you for the update and for taking the time to read our rebuttal. We appreciate the thoughtful review and your helpful suggestions. Accordingly, we will carefully revise the manuscript.

---

### Official Review · Reviewer_2rkk · 2026-03-09

**Soundness:** 3
**Presentation:** 3
**Significance:** 3
**Originality:** 2
**Overall Recommendation:** 4
**Confidence:** 3

**Summary:**

This paper proposes Turbo4DGen, an acceleration framework for diffusion-based 4D generation that reduces the computational cost of the spatial–camera–motion attention mechanism at token, block, and chain levels. Experiments demonstrate that Turbo4DGen achieves up to 9.7× inference speedup and reduced memory usage while maintaining comparable generation quality.

**Compliance With Llm Reviewing Policy:**

Affirmed.

**Final Justification:**

They have addressed my original concerns very well, and they also conducted additional experiments on the newly raised questions to further resolve my concerns.

**Key Questions For Authors:**

1.The paper compares against DeepCache and AT-EDM. Since these methods are extended by the authors to the 4D setting, could the authors provide more implementation details on how these adaptations were performed and how hyperparameters were tuned?

2.The proposed method is fine-tuned on a subset of the ObjaverseDy dataset. To what extent does this fine-tuning contribute to the improved generation quality reported in Table 1 compared to the SV4D baseline?

3.The SCM attention chain is originally designed to enforce spatial, temporal, and multi-view consistency simultaneously. Since Turbo4DGen introduces both token pruning and chain-level bypassing, could the authors provide additional analysis on how these operations affect consistency across views and frames?

**Limitations:**

yes

**Strengths And Weaknesses:**

Strengths:

1.The paper addresses the computational inefficiency of diffusion-based 4D generation and proposes a multi-level acceleration framework targeting the spatial–camera–motion (SCM) attention mechanism.

2.If the reported speedups and quality preservation are robust across settings, the proposed framework could serve as a useful systems-level improvement for future 4D generation models. The impact of the work may depend on how broadly the method can generalize beyond the specific SV4D-style SCM architecture.

3.While the components themselves are not entirely new, the multi-scale integration of these techniques and their application to the SCM attention structure constitute a reasonably original systems-level contribution.

Weaknesses:

1.The comparisons with prior acceleration methods (e.g., DeepCache and AT-EDM) rely on adaptations of these methods to the 4D setting implemented by the authors, and the details of these adaptations are not fully described, which makes it difficult to assess whether the comparisons are fully fair.

2.The method is fine-tuned on the ObjaverseDy dataset, which raises some uncertainty about how much of the improved quality comes from the acceleration framework itself versus additional task-specific adaptation.

3.The citation style could be improved for readability. In multiple paragraphs, the same references are repeatedly cited within very short spans of text, even when discussing the same line of work. Consolidating such repeated citations or grouping them at the end of sentences would make the narrative smoother and improve the overall reading experience.

---

> ### Author Rebuttal · Authors · 2026-03-31
>
> We sincerely thank you for the constructive feedback and suggestions.
>
> ### Implementation and Adaptation of Other Methods
>
> Thanks for pointing it out. Due to the page limit, we explain the implementation details in Appendix B, explaining how these 2D/3D acceleration methods were extended to the 4D SCM chain. In summary, we explain the code adaptation as follows:
>
> - **DeepCache Adaptation**: Originally designed for text-to-image U-Nets, we carefully map the core caching logic (pushing cache at the final upsampling block and reusing it in the subsequent denoising step at the same final upsampling block while skipping preceding blocks) from its U-Net block to the 4D generation diffusion block. However, we observe that this base-level 2D cache is insufficient for 4D generation, as it fails to maintain the complex spatio-temporal and view consistency required by the SCM chain.
>
> - **AT-EDM Adaptation**: We integrate its important G-WPR token-selection algorithm for independent attention pruning within the SCM chain. Crucially, to isolate the impact of our proposed semantic-aware approach, this baseline adaptation does not utilize a rolling cache refilling mechanism and performs pruning independently across SCM layers. This comparison demonstrates that without the integrated semantic guidance and spatiotemporal caching of Turbo4DGen, standard 2D/3D pruning methods struggle to maintain the same efficiency-quality balance in 4D generation.
>
> ### Fine-Tuning
>
> We apologize for the insufficient explanations and highlights in the Experiment section. Our evaluations consist of both fine-tuning and no fine-tuning. (1) Table 1 shows the evaluation on ObjaverseDy with fine-tuning. The fine-tuning on a subset of ObjaverseDy was primarily intended to recover any potential fidelity loss introduced by the structural changes of token pruning and caching. (2) Table 2 shows the evaluation on Consistent4D without fine-tuning. Consistent4D is a widely used dataset in the era of 4D generation and has become a benchmark for existing 4D generation research. However, it only releases the ground truth for the test set, which is a very appropriate testbed for evaluating the zero-shot ability of our method.
>
> We summarize additional results on ObjaverseDy without fine-tuning in Table R3. This experiment shows that our acceleration modules alone achieve significant speedups with only marginal quality trade-offs compared to the original SV4D. This confirms that the efficiency gains are an architectural benefit of our Turbo4DGen. We will revise the manuscript accordingly and include this table.
>
> **Table R3.** Evaluation (without fine-tuning) on ObjaverseDy.
> | Method | Speedup | LPIPS | PSNR | CLIP-S | SSIM |
> |---|---|---|---|---|---|
> | SV4D | 1.0 | 0.122 | 18.47 | 0.902 | 0.884 |
> | Turbo4DGen (Ours) | 5.8 | 0.125 | 18.36 | 0.891 | 0.881 |
>
> ### Consistency Analysis
>
> Due to rebuttal time limits, we evaluate generated multi-view images by reporting FVD-based metrics in Table 10, which are widely used to measure consistency. We use the same metrics as baseline to ensure a fair comparison, e.g., FVD-F (calculate FVD over frames as each view), FVD-V (calculate FVD over views at each frame), FVD-Diag (calculate FVD over the diagonal images of the image matrix), and Fv4D (calculate FVD over all images by scanning them in a bidirectional raster order).
>
> As shown in Table R4, pruning slightly affects spatial fidelity (FVD-F), caching mainly helps maintain it, and bypass mainly affects cross-view/diag fidelity (FVD-Diag), illustrating how each strategy differently impacts the model’s quality.
>
> Additionally, we further generate 4D videos to illustrate the visual consistency of our method on the iPhone dataset [R2]. Those example videos can be found at [Anonymous Github](https://anonymous.4open.science/r/anon-code-x9j2/). We will provide more videos and interactive demos on our project website in the future.
>
> **Table R4.** Evaluation of 4D outputs on ObjaverseDy dataset. Turbo4DGen can maintain performance in both video frames and multi-view consistency.
>
> | Model | LPIPS↓ | CLIP-S↑ | FVD-F↓ | FVD-V↓ | FVD-Diag↓ | FV4D↓ |
> |---|---|---|---|---|---|---|
> | SV4D | 0.122 | 0.902 | 754.23 | 436.86 | 666.59 | 699.04 |
> | Turbo4DGen w/o pruning | 0.112 | 0.920 | 618.52 | 409.15 | 640.11 | 545.19 |
> | Turbo4DGen w/o caching | 0.115 | 0.914 | 640.79 | 415.32 | 645.48 | 555.62 |
> | Turbo4DGen w/o bypass | 0.111 | 0.919 | 622.50 | 409.07 | 638.82 | 547.20 |
> | Turbo4DGen (Ours) | 0.113 | 0.917 | 626.36 | 410.93 | 642.26 | 549.03 |
>
> ### Citation Style
>
> We sincerely thank the reviewer for this suggestion regarding our citation style. We have carefully revised the manuscript to consolidate references and place them at the end of relevant sentences.
>
> [R2] Hang Gao et al., Monocular Dynamic View Synthesis: A Reality Check, NeurIPS2022.

---

> > ### Author Rebuttal · Reviewer_2rkk · 2026-04-02
> >
> > Thank you for the response and additional experiments! I will maintain the score(4: Weak accept).

---

> > > ### Author Response · Authors · 2026-04-05
> > >
> > > Thank you for the update and for taking the time to read our rebuttal. We appreciate the thoughtful review and your helpful suggestions. Accordingly, we will carefully revise the manuscript.

---

### Official Review · Reviewer_VL3B · 2026-03-11

**Soundness:** 3
**Presentation:** 3
**Significance:** 3
**Originality:** 3
**Overall Recommendation:** 4
**Confidence:** 4

**Summary:**

This work focuses on the computational cost of 4D generation task (OOM and long generation time). Specifically, this work proposes Turbo4DGen, which introduces the spatiotemporal cache mechanism for reusing the intermediate attention across denoising steps. Experimental results show that the proposed Turbo4DGen achieves an average 9.7× speedup.

**Compliance With Llm Reviewing Policy:**

Affirmed.

**Final Justification:**

My concerns have been adequately addressed. The authors provide further experimental results. I have no further questions. I keep my score of 4.

**Key Questions For Authors:**

* I have no further questions. Please focus on the issue of the selection of the base 4D generation model mentioned in the Weakness.

**Limitations:**

I did not find the Impact Statement, and the declaration of the limitations of this work.

**Strengths And Weaknesses:**

**Strength**

* This work is well-written, and the readers can understand the proposed points.
* The overall speedup ratio and the comparable generation quality show the effectiveness of this work.
* The motivation part and the evaluation part are abundant.

**Weakness**

* The main concern of this work is the selected base model. I notice that the overall illustration and evaluation are based on the SV4D. I wonder that if the proposed method is transferable and can be applied to other 4D generation models, or if the proposed method is more of an optimization of SV4D. I recommend that the authors provide more experiments and analyses to expand the scope of their method.
* I think the ablation is one of the core experiments, and I suggest adjusting the article layout to place the ablations in the main manuscript rather than in the appendix.
* I did not find the Impact Statement, and the declaration of the limitations of this work. Please supplement them.

---

> ### Author Rebuttal · Authors · 2026-03-31
>
> We sincerely thank you for the constructive feedback.
> ###  Baseline Model and Transferability
> Thanks for pointing it out. Turbo4DGen is to systematically accelerate 4D generation models, the former multi-view 4D content generation, instead of the latter 4D reconstruction or rendering. We have to select SV4D as our baseline because, to the best of our knowledge, it was the **only** state-of-the-art open-source 4D generation model. Other state-of-the-art works, e.g., CAT4D and 4Real, remain closed source (model, dataset, etc.), which prevents us from accelerating or building upon them.
>
> Our Turbo4DGen is a general optimization for accelerating the Spatio-Camera-Motion (SCM) attention mechanism in 4D generation, and our core multi-level computation skipping is designed as general plug-and-play components. For other 4D generation models, e.g., CAT4D and 4Real, which follow a similar attention mechanism, we believe Turbo4DGen can be easily employed to accelerate them.
>
> Due to rebuttal time limits, we tried our best to apply Turbo4DGen to accelerate TrajectoryCrafter [R1], a notable model leveraging a similar SCM paradigm. The bottleneck is its SCM-like attention chain, which consists of a dual-stream conditional video diffusion process including Ref-DiT (cross-attention monocular input and camera-trajectory point clouds) and VideoDiT. Following the protocol in TrajectoryCrafter [R1], we evaluate our Turbo4DGen on the iPhone [R2] dataset (in-the-wild) without re-tuning. As shown in the following Table R2, our method transfers effectively to alternative baselines while yielding stable performance. Those results can be found at [Anonymous Github](https://anonymous.4open.science/r/anon-code-x9j2/). We will open-source our method and test more state-of-the-art models once they become public.
>
> **Table R2**. Quantitative comparison of novel trajectory video synthesis on the iPhone [R2] dataset. The pruning rate is set as 0.5.
> | Method | Scene | Speedup | PSNR (dB) | SSIM | LPIPS |
> |---|---|---|---|---|---|
> | TrajectoryCraft [R1] | Apple | 1.00 | 13.88 | 0.285 | 0.612 |
> | | Block | 1.00 | 14.21 | 0.528 | 0.479 |
> | | Paper | 1.00 | 14.89 | 0.482 | 0.471 |
> | | Spin | 1.00 | 14.51 | 0.380 | 0.518 |
> | | Teddy | 1.00 | 13.73 | 0.411 | 0.513 |
> | | *Mean* | *1.00* | *14.24* | *0.417* | *0.519* |
> | Ours | Apple | 4.93 | 13.62 | 0.263 | 0.601 |
> | | Block | 5.14 | 13.96 | 0.506 | 0.467 |
> | | Paper | 4.70 | 14.65 | 0.461 | 0.458 |
> | | Spin | 5.20 | 14.24 | 0.359 | 0.507 |
> | | Teddy | 4.80 | 13.47 | 0.389 | 0.503 |
> | | *Mean* | *4.95* | *14.00* | *0.396* | *0.507* |
>
> ### Manuscript Layout
> We appreciate the suggestion to move the ablation studies to the main manuscript to highlight their significance. We will reorganize the manuscript to move the key ablation results from the Appendix into Section 5.2.
>
> ### Impact Statement & Limitations
> Thanks for pointing it out. We will add a section to discuss the limitations and a section for the impact statement in the revised manuscript.
> - **Limitations:** Our framework is optimized for standard 4D generation settings (≥20 denoising steps) and SCM-based architectures, where it demonstrates robust performance. However, its effectiveness may degrade in very low-step settings or in scenarios with rapid object motion (e.g., rotation), where reduced $V_{ASR}$ makes it harder to trigger the bypass threshold and maintain efficiency–consistency trade-offs.
> - **Community Impact:** We emphasize that Turbo4DGen democratizes 4D generation by enabling high-fidelity results on consumer-grade hardware, such as the NVIDIA RTX PRO 6000. This lowers the barrier for researchers who do not have access to industrial-scale compute clusters.
>
> [R1] Mark Yu et al., TrajectoryCrafter: Redirecting Camera Trajectory for Monocular Videos via Diffusion Models, ICCV 2025 (Oral).
>
> [R2] Hang Gao et al., Monocular Dynamic View Synthesis: A Reality Check, NeurIPS2022.
>
> We hope these revisions and the clarification on the open source landscape strengthen the paper and address your concerns.

---

> > ### Author Rebuttal · Reviewer_VL3B · 2026-04-02
> >
> > I keep my score.

---

> > > ### Author Response · Authors · 2026-04-05
> > >
> > > Thank you for the update and for taking the time to read our rebuttal. We appreciate the thoughtful review and your helpful suggestions. Accordingly, we will carefully revise the manuscript.

---

### Official Review · Reviewer_EyCH · 2026-03-12

**Soundness:** 2
**Presentation:** 2
**Significance:** 2
**Originality:** 2
**Overall Recommendation:** 4
**Confidence:** 3

**Summary:**

This paper contributes a practical acceleration framework for diffusion-based 4D generation. To reduce the computation overhead from SCM attention, the paper proposes a three-level acceleration pipeline:

- block-level: rolling-cache reuse across denoising steps,
- token-level: semantic-aware token pruning,
- chain-level: an adaptive chain-bypass scheduler

**Compliance With Llm Reviewing Policy:**

Affirmed.

**Final Justification:**

My major concerns are solved, thus, I am raising my score to Weak accept.

It would strengthen the paper by adding a limitation section.

**Key Questions For Authors:**

1. For a fair comparison, are the speedups measured under exactly matched inference settings, including denoising steps, sampler?
2. Are the chosen thresholds and cache policy generalize across datasets or resolutions without retuning?

**Limitations:**

I do not see a dedicated Limitations section. It would strengthen the paper by clearly discussing failure modes, robustness boundaries, generalization limits

**Strengths And Weaknesses:**

## Strengths
1. **Well-motivated practical problem**. The paper targets an important bottleneck in 4D generation: inference efficiency. This is a meaningful systems problem because practical deployment is still limited by heavy multi-view/multi-frame diffusion inference.
2. **Reasonably clear method decomposition**. The token-level, block-level, and chain-level design is easy to follow.
3. **Comprehensive evaluation**. The paper appears to include both quantitative and qualitative results, plus ablations on pruning, bypass thresholds, and strategy variants.
4. **Strong practical speedup**. The results demonstrate that the reported acceleration is substantial enough to matter in practice.

## Weaknesses:
1. Robustness of heuristic scheduler: The cache similarity metric and bypass thresholding are plausible but under-analyzed. The paper should provide more evidence for robustness, threshold transferability, and failure modes.

---

> ### Author Rebuttal · Authors · 2026-03-31
>
> We sincerely appreciate your constructive feedback and your recognition of strengths, such as the practical significance of our speedup.
>
> ### Motivation of Adaptive Scheduler
> As illustrated in Figure 7 of the manuscript, we provide a detailed analysis using three randomly selected samples to visualize the relationship between $V_{ASR}$ and feature similarity. This analysis indicates that the $V_{ASR}$ metric serves as a reliable predictive proxy, allowing the scheduler to anticipate future cache similarity and bypass redundant SCM computations without compromising temporal or view consistency.
>
> ### Robustness of Scheduler
>
> The thresholds for $V_{ASR}$ and the cache policy remain constant across all datasets (i.e., ObjaverseDy, Consistent4D. We have conducted additional experiments t to show the robustness of our scheduler, as summarized in Table R1. Together with Table 5 (Appendix E.4), the robustness is reflected in the following aspects:
>
>  - **Zero-shot Robustness without re-tuninig**: The evaluation  (Table 2) on Consistent4D is entirely zero-shot **without re-tuning**, as this dataset only provides a testing set. The results show that our method consistently achieves the highest performance without any re-tuning, demonstrating the strong generalization of our scheduler.
>
>  - **Resolution insensitivity**: We test both the default 576×576 and the higher 768×768 (OOM in the baseline) resolution, demonstrating that our scheduler maintains stable performance across all datasets. Higher resolution yields a slight improvement but requires exponential computing resources, and our approach effectively manages this trade-off.
>
> - **Effective threshold**: Evaluated on varying datasets and settings, our scheduler with $V_{ASR}$ maintains constant performance, e.g., speedup and generation quality. And results indicate that the 0.9 (default) setting manages the trade-off. Note that we allow different formulations for $V_{ASR}$ to handle various complex scenarios (we included in the revised version).
>
> **Table R1**. Performance analysis with different scheduler settings on ObjaverseDy and Consistent4D datasets.
> | Denoising Step | Resolution | Threshold | ObjaverseDy        | | | Consistent4D (w/o Tuning) | | |
> |---|---|---|---|---|---|---|---|---|
> | | | | Speedup | PSNR | CLIP-S | Speedup | PSNR | CLIP-S |
> | 20 | 576 × 576 | 0.85 | 11.73 | 17.81 | 0.863 | 11.54 | 17.09 | 0.889 |
> | | | 0.90 | 9.70 | 20.27 | 0.917 | 9.67 | 18.25 | 0.927 |
> | | | 0.95 | 9.56 | 20.29 | 0.919 | 9.42 | 18.28 | 0.928 |
> | | 768 × 768 | 0.85 | 11.98 | 17.46 | 0.866 | 11.63 | 16.92 | 0.893 |
> | | | 0.90 | 9.81 | 20.21 | 0.920 | 9.72 | 18.18 | 0.933 |
> | | | 0.95 | 9.63 | 20.25 | 0.924 | 9.47 | 18.19 | 0.935 |
> | 30 | 576 × 576 | 0.85 | 13.52 | 19.11 | 0.880 | 12.96 | 18.87 | 0.904 |
> | | | 0.90 | 10.63 | 21.08 | 0.922 | 10.41 | 19.07 | 0.929 |
> | | | 0.95 | 10.41 | 21.13 | 0.925 | 10.30 | 19.09 | 0.929 |
> | | 768 × 768 | 0.85 | 13.66 | 18.83 | 0.883 | 13.03 | 18.81 | 0.909 |
> | | | 0.90 | 10.72 | 21.01 | 0.924 | 10.51 | 19.02 | 0.930 |
> | | | 0.95 | 10.67 | 21.05 | 0.925 | 10.44 | 19.03 | 0.931 |
>
> ### Matched Inference Settings
>
>  For a fair comparison, all reported speedups are measured under identical inference settings, as described in Section 5. Specifically, 20 denoising steps, EDM sampler (Karras et al., 2022), camera poses, number of frames and views, and resolution. Furthermore, all experiments were conducted on the same NVIDIA RTX PRO 6000 GPUs.
>
> ### Limitation Section
>
> We thank the reviewer for pointing this out. We will add a section to discuss the limitations in the revised manuscript.
>
> Our framework is optimized for standard 4D generation settings (≥20 denoising steps) and SCM-based architectures, where it demonstrates robust performance. However, its effectiveness may degrade in very low-step settings or in scenarios with rapid object motion (e.g., rotation), where reduced $V_{ASR}$ makes it harder to trigger the bypass threshold and maintain efficiency–consistency trade-offs.
>
> We hope these clarifications and the added discussion address your concerns.

---

> > ### Author Rebuttal · Reviewer_EyCH · 2026-04-05
> >
> > Thanks for your detailed rebuttal.
> >
> > My major concerns are solved, thus, I am raising my score to Weak accept.
> >
> >  It would strengthen the paper by adding a limitation section.

---

> > > ### Author Response · Authors · 2026-04-06
> > >
> > > Thank you for the update and for taking the time to read our rebuttal. We appreciate the thoughtful review and your helpful suggestions. Accordingly, we will carefully revise the manuscript.

---

### Decision · Program_Chairs · 2026-04-30

**Decision:**

Accept (regular)

**Comment:**

The paper received four weak accepts from the reviewers. The reviewers acknowledged the technical contribution of the paper, the strong qualitative evaluation in terms of speedup performance, and the presentation quality. Overall, the reviewers were all positive about this submission, although they did not feel that the paper was completely new in terms of some designs in the proposed model. Based on the overall rating distribution and the rebuttal provided by the authors, AC decided to recommend a weak accept for this submission.